# Ecophysiological Plasticity and Cold Stress Adaptation in Himalayan Alpine Herbs: *Bistorta affinis* and *Sibbaldia procumbens*

**DOI:** 10.3390/plants8100378

**Published:** 2019-09-27

**Authors:** Inayat Ur Rahman, Robbie Hart, Aftab Afzal, Zafar Iqbal, Abdulaziz A. Alqarawi, Elsayed Fathi Abd_Allah, Abeer Hashem, Farhana Ijaz, Niaz Ali, Eduardo Soares Calixto

**Affiliations:** 1Department of Botany, Hazara University, Mansehra-21300, KP, Pakistan; zafariqbal@hu.edu.pk (Z.I.); fbotany@yahoo.com (F.I.); niaz@hu.edu.pk (N.A.); 2William L. Brown Center, Missouri Botanical Garden, 4344 Shaw Blvd., P.O. Box 299, St. Louis, MO 63110, USA; Robbie.hart@mobot.org; 3Department of Plant Production, College of Food and Agriculture Science, King Saud University, Riyadh 11451, Saudi Arabia; alqarawi@ksu.edu.sa (A.A.A.);; 4Botany and Microbiology Department, College of Science, King Saud University, Riyadh 11451, Saudi Arabia; habeer@ksu.edu.sa; 5Mycology and Plant Disease Survey Department, Plant Pathology Research Institute, Agriculture Research Center, Giza 12619, Egypt; 6Department of Biology, University of Sao Paolo, SP 05315-970, Brazil; calixtos.edu@usp.br; 7Department of Biology, University of Missouri, St. Louis, MO 63166-0299, USA

**Keywords:** Ecophysiology, phytohormones, elevation, temperature, cold stress, Himalayas

## Abstract

Plants have evolved several metabolic pathways as a response to environmental stressors such as low temperatures. In this perspective, it is paramount to highlight physiological mechanisms of plant responses to altitudinal gradients as a proxy to evaluate changing environments. Here, we aimed to determine the impact of elevation on the physiological attributes of two plant species along an altitudinal gradient. Our hypothesis was that the altitudinal gradient influences proline, protein, and sugar contents, as well as abscisic acid (ABA) and indole acetic acid (IAA) concentrations. We studied these physiological components in leaves collected from four different altitudinal ranges in Himalayan region of Pakistan from two native herbs, namely *Bistorta affinis* and *Sibbaldia procumbens*. Leaves were collected at the initial blooming phase from each altitudinal range, viz. 2850 m, 3250 m, 3750 m and 4250 m. We observed that most abiotic factors decrease with altitude which induces cold acclimation. A significant increase in the concentration of physiological components was observed as altitude increased, except for IAA, which decreased. Furthermore, we did not find variations in proline, ABA and IAA concentrations between species; only sugar and protein, with higher values for *B. affinis.* We conclude that altitudinal gradients significantly affect the physiological components of *B. affinis* and *S. procumbens* in Himalayan region. This result contributes to the understanding of how plants adapt to environmental pressures, acting as a proxy for the evaluation of impacts caused by climate changes.

## 1. Introduction

Plants experience cold or chilling stress at low temperatures from 0–15 °C. Under such circumstances, plants try to uphold homeostasis to attain freezing tolerance which involves wide-ranging reprograming of gene expression and metabolism [1]. Plants adapted to low temperature stress involve changes in several metabolic pathways, including carbohydrates synthesis [2]. In many plants, sucrose is accumulated during cold stress [3]. During cold stress, most of the tolerant plant species produce a new set of proteins that is correlated with the increase of cold hardiness. Altitudinal variation also has a significant effect on proline contents of medicinal plant leaves [4]. Different experiments have shown that besides other solutes, the level of free amino acids, especially proline, increased during cold hardening [5].

Phytohormones are known to play a crucial role in plant growth and development in response to environmental stress [6]. For instance, abscisic acid (ABA) is one of the stress hormones that plays a critical role in regulating plant water status, osmotic stress tolerance [7] and adaptation to abiotic stresses [8]. ABA can act as a long-distance communication signal between water deficit roots and leaves by inducing the closure of stomata and reducing water loss through transpiration [9]. Fahad and Bano [10] stated that variations in altitude had a significant effect on endogenous levels of ABA, indole acetic acid (IAA) and Gibberellic acid concentrations of plant leaves.

Altitudinal gradients are excellent tools for analyzing plant responses to stress and plant adaptation, especially considering abiotic factors. They may also be used as proxy for understanding the effects of climate change on the adaptive and survival potential of living beings, especially plants. By analyzing these behavioral patterns in plants along elevational gradients, we indirectly measure possible impacts that climate change may have on particular environments.

Regarding protein, sugar, and proline content and phytohormone variation according to altitudinal gradients, the following question arises: does plant ecophysiology play an important role in assessing the response of plants to global change, considering the altitudinal gradient as a proxy? We aimed to determine the relationship of physiological variations in two plants species along the altitudinal gradient, based on the hypothesis that the altitudinal gradient influences protein, sugar and proline contents and phytohormone concentrations, in which some physiological components increase their concentrations, while others decrease.

## 2. Results

### 2.1. Ecophysiological Plasticity in *Bistorta affinis*

Our results showed that all analyzed components (proline, protein, sugar, ABA, IAA) showed significant differences among the observed altitudes (Table 1). Maximum proline (5.78 ± 0.68 mg/g.dwt; mean ± SD), protein (1.55 ± 0.06 mg/g.dwt), sugar (8.50 ± 0.45 mg/g.dwt), and ABA (534.67 ± 33.9 ηg/g.dwt) concentrations were found at highest elevational range (4250 m) (Table 2; Figure 1). The highest amount of IAA content (414.00 ± 17.9 ηg/g.dwt) was recorded in the sample collected from the lowest altitudinal range (2850 m) (Table 2; Figure 1). On the other hand, only protein and sugar presented a difference in concentration between species (Table 1); *B. affinis* presented higher concentration of both compounds (Figure 1).

### 2.2. Ecophysiological Plasticity in *Sibbaldia procumbens*

As observed, our results showed that height and species influenced almost all components analyzed (Table 1). Maximum proline (6.58 ± 0.52 mg/g.dwt; mean ± SD), protein (1.42 ± 0.1 mg/g.dwt), sugar (7.41 ± 0.38 mg/g.dwt), and ABA (482.00 ± 16.87 ηg/g.dwt) concentrations were found at highest elevational range (4250 m) (Table 3; Figure 1). The highest amount of IAA content (406.67 ± 16.1 ηg/g.dwt) was recorded in the sample collected from the lowest altitudinal range (2850 m) (Table 3; Figure 1). Only protein and sugar presented difference in concentration between species (Table 1); *S. procumbens* presented lower concentration of both compounds (Table 3; Figure 1).

### 2.3. Non-Metric Multidimensional Scaling (NMDS)

The altitudes analyzed showed a significant difference through the environmental factors evaluated (NMDS stress = 0.00, ANOSIM *p* < 0.001; Table 4). In addition, the specific altitudes showed variation in the concentration of plant compounds (NMDS stress = 0.031, ANOSIM *p* < 0.001) (Figure 2a), which was not seen when we compared plant species (NMDS stress = 0.031, ANOSIM *p* = 0.934) (Figure 2b).

## 3. Discussion

Our results evidenced that altitudinal gradients significantly affecting the physiological attributes of *Bistorta affinis* and *Sibbaldia procumbens* in Himalayan region corroborating our main hypothesis. As altitude increases, environmental variables vary, and plants need to adapt to increasing environmental stress. In general, we observed an increase in proline, protein, sugar, and ABA concentrations, and a decrease in IAA, as the altitude increases. It is known that altitude represents a complex gradient along which many environmental factors change concomitantly [11]. Precipitation, light intensity and radiation intensity increases [12,13] and temperature decreases [14] with rising altitude. Therefore, the impact of altitude on plant growth is the result of the combined action of various factors [13].

Our results explicitly showed significant variations in proline contents of leaves collected from different altitudinal ranges, with maximum proline content value at the highest elevation (4250 m). Accumulation of proline is an important abiotic stress indicator in higher plants [15] which may involve osmoregulation and acts as a cellular osmotic regulator or radical scavenger [16,17]. Ashraf and Harris [18] studied a strong positive correlation between proline accumulation and plant adaptation. The analytical results revealed that the proline content marked significant increase at highest altitude and low temporal ranges which reflects the physiological disturbance in both understory species. The temperature decreases with increase in altitude which makes unfavorable environmental conditions. Many researchers clearly reported that besides other solutes, the level of proline significantly increased during cold hardening [19].

The protein content in leaves of both species (i.e., *Sibbaldia procumbens* and *Bistorta affinis*) was found significantly maximum with increase in the altitude (2850–4250 m). In *B. affinis*, higher protein content was found on same altitude in comparison with other plant species S. *procumbens* (1.55 mg/g.dwt). Protein content of leaves with respect to altitudinal variation was studied in many plant species, and it was observed that during acclimation of cold, most freezing tolerant plant species produce new set of proteins that was correlated with the increase of cold hardiness [4]. Thus, we can see that production of new proteins or higher concentrations as seen in our study is a result of an adaptation of plants to environmental pressures.

In the case of sugar content, highly significant variations were recorded among the collected leaves sampled from different altitudinal ranges with maximum sugar content records at highest elevation (4250 m). Plants adapted to low temperature stress involve changes in several metabolic pathways including carbohydrates synthesis [2], which plays an important role in plant stress tolerance [20] and increasing sugar content acts as osmoregulant [21]. In many plants, sucrose was accumulated during cold stress [3]. For instance, Bano et al. [20] noticed maximum level of sugar content in the higher altitude alpine herbs of Hunza Valley, Pakistan, similar to what was found in our study.

The results indicated highly significant differences in ABA content of leaves collected from different altitudinal ranges, with the highest amount of ABA collected from the highest altitudinal range (4250 m). Similarly, higher endogenous ABA content was collected from 4350 m compared to ABA collected from 1650 m [10]. ABA accumulates in plants with response to a range of environmental stresses, including low temperature and water stress, and studies have demonstrated this adaption of plants [22]. Thus, ABA induction appears to aid in osmotic regulation and acts as a plant response to water stress environments.

In the other hand, IAA contents was found significantly increased in plant leaves collected from lower altitudinal ranges (2850 m) compared to higher collected peaks (4250 m). IAA is a growth hormone that promotes differential cell elongation and functions as a plant growth regulator. During stress reactions, it is known that the concentration of phytohormones-inhibitors increases [23] and this increase may cause variations in the plant fitness or growth. Changes in the hormonal balance were thought to be responsible for the cessation of plant growth in the cold [24]. From this perspective, we point out that due to higher environmental pressures at higher altitudes phytohormones-inhibitors concentrations increase and consequently decrease the concentration of IAA.

## 4. Materials and Methods

### 4.1. Study Site, Experimental Design and Environmental Gradients

This experiment was undertaken in Manoor Valley, Himalayan region of Pakistan. The valley is floristically rich and diverse, but little explored and studied. It is characterized by a harsh climate and strong seasonality and presents different vegetational zones. Despite this flora richness, the few studies done in the region are related to herbal reports or preliminary checklists. Four different altitudinal ranges were selected from moist temperate areas to alpine pastures (based on the species abundance i.e., 2850 m, 3250 m, 3750 m and 4250 m). Clinometer and geographical positioning system (GPS) were used at each sampling site to record aspect and elevation respectively. For other environmental gradients (i.e., barometric pressure, dew point, humidity, heat index, temperature, wet bulb and wind speed) were determined using handheld weather station (Kestrel weather tracker 4000). Two hundred grams of soil samples from each sampling site were collected to determine the soil texture, pH, electric conductivity (EC), organic matter (OM), calcium carbonate (CaCO_3_), potassium and phosphorous (Table 4).

### 4.2. Investigational Species, Sample Collection and Physiological Attributes

The impact of altitudinal gradient on the physiological attributes of high-altitude native plant species was determined. The understory investigational plant species were *Bistorta affinis* (Polygonaceae) and *Sibbaldia procumbens* (Rosaceae). *Bistorta affinis* has elliptic leaves covered abaxially by a layer of waxy coating, which adds a whitish color to the leaf. Flowers are densely clustered in spike inflorescences. *Sibbaldia procumbens* has three-leaflet leaves that are 5–10 cm long. Flowers present bractlets, green sepals and yellowish petals, which are too small (~1 mm).

Fresh leaves samples were collected from each sampling site in blooming season of both plants during 2017. Expanded and intact flag leaves of similar size in the bloom phase were collected and replicated three times at each sampling site. Then, the leaves were wrapped in aluminum foil, immediately immersed in liquid nitrogen, and stored at −80 °C for laboratory analysis. These samples were further investigated for protein, sugar and proline contents, abscisic acid (ABA) and indole acetic acid (IAA). Protein content of leaves was determined by following the method of Lowery et al. [25] using bovine serum albumin (BSA) as standard. Sugar estimation of fresh leaves was determined by following method of Dubois et al. [26]. The proline contents of leaves were measured by the method of Bates et al. [27]. The ABA and IAA concentration were determined by the method of Kettner and Doerffling [28].

### 4.3. Statistical Analyses

After laboratory investigations, all the data regarding physiological attributes were entered into an MS-Excel spreadsheet for computational work. We used a Non-Metric Multidimensional Scaling (NMDS) followed by Analysis of Similarities (ANOSIM) and Generalized Linear Models (GLM) with Gaussian distribution followed by Anova from package “car” [29] to verify if the concentration of plant compounds (proline, protein, sugar, ABA, and IAA) varies between altitudes, and between species. Post-hoc test was done with least significant differences (LSDs) among the means. In GLM plant compounds were considered as response variable, while altitude and plant species were considered as predictors. We also used NMDS and ANOSIM to compare environmental factors among altitudes. NMDS and ANOSIM were conducted in R 3.5.1 software using the package “vegan” [30], the Bray–Curtis dissimilarity index, and 999 permutations. 

### 4.4. Ethical Approval

This study was approved by the “Board of Study members of the Department of Botany” as well as “Advanced Studies and Research Board, Hazara University Mansehra, Pakistan”.

## 5. Conclusions

The current investigation supports evidence that altitudinal gradients significantly affect the physiological attributes of *Bistorta affinis* and *Sibbaldia procumbens* in the Himalayan region. As altitude increases, environmental variables vary, and plants need to adapt to this environmental stress. In this study, we showed that both plant species studied increase proline, protein, sugar, and ABA concentrations, and decrease IAA, as the altitude increases. This result contributes to the understanding of how plants adapt to environmental pressures, acting as a proxy for the analysis of impacts caused by climate changes. Thus, we suggest that studies be done to evaluate the different plant compounds that can help in understanding the adaptation of plants in hostile environments and use the results of these analyzes as ways of preserving and mitigating the impacts caused by climate change.

## Figures and Tables

**Figure 1 plants-08-00378-f001:**
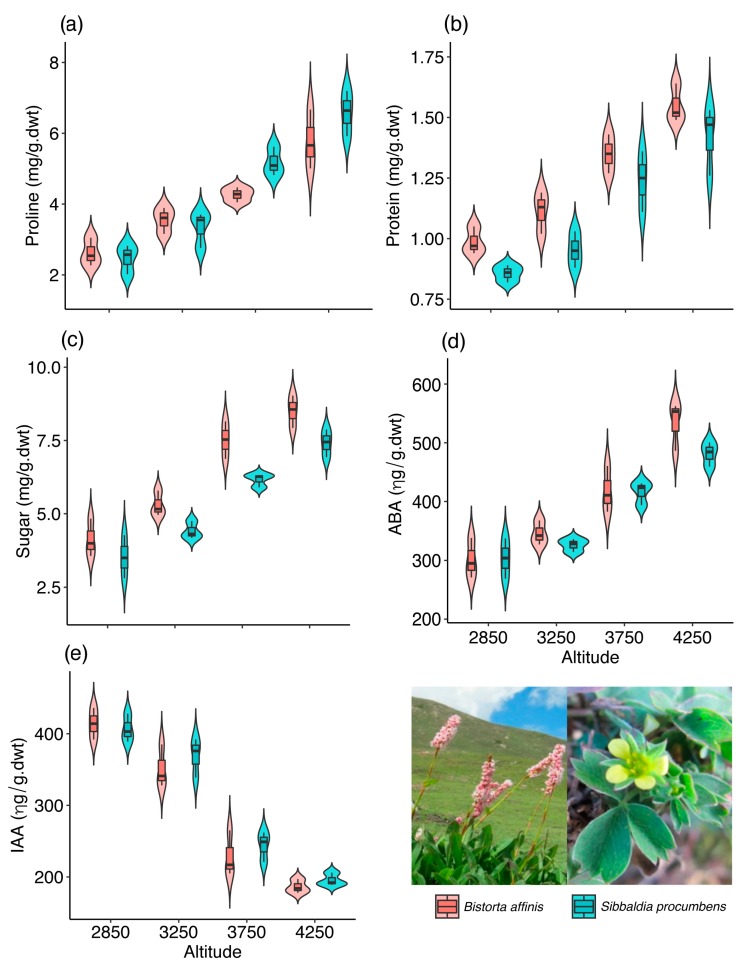
Concentration of ABA (**a**), IAA (**b**), proline (**c**), protein (**d**), and sugar (**e**) in *Bistorta affinis* and *Sibbaldia procumbens* in different altitudes. Figures show a violin plot represented by a boxplot and a rotated kernel density (probability density) plot. GLM results for each plant compound are in Table 1.

**Figure 2 plants-08-00378-f002:**
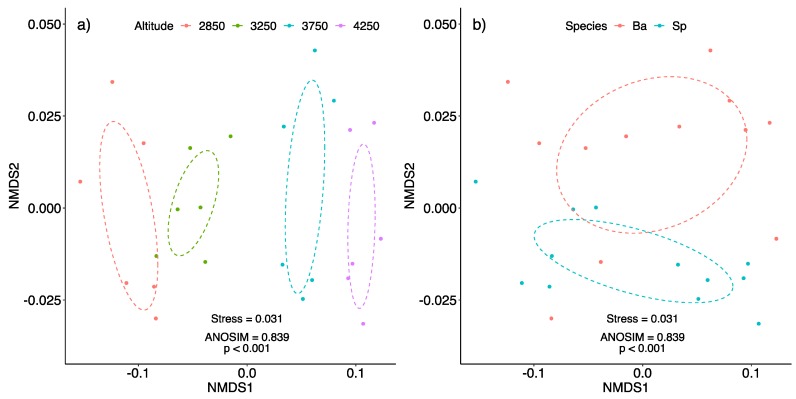
Non-Metric Multidimensional Scaling (NMDS) with 95% confidence ellipse obtained from the plant compounds analyzed (proline, protein, sugar, ABA, IAA) at different altitudes among (**a**) sampling sites and (**b**) species. The analyzed altitudes differed significantly in relation to the analyzed plant compounds, differently from plant species.

**Table 1 plants-08-00378-t001:** Generalized Linear Model (GLM) of plant compounds concentration between plant species and altitude.

Compound	Factor	χ^2^	df	*p*-Value
**ABA**	Altitude	173.00	3	<0.001
	Species	2.23	1	0.1353
**IAA**	Altitude	409.57	3	<0.001
	Species	0.97	1	0.3246
**Proline**	Altitude	149.944	3	<0.001
	Species	2.273	1	0.1316
**Protein**	Altitude	165.446	3	<0.001
	Species	14.938	1	<0.001
**Sugar**	Altitude	245.805	3	<0.001
	Species	22.658	1	<0.001

**Table 2 plants-08-00378-t002:** Ecophysiological plasticity observed in *Bistorta affinis* through the analysis of proline, protein, and sugar contents, and abscisic acid (ABA) and indole acetic acid (IAA) concentrations.

Altitude (m)	Proline Content (mg/g.dwt)	Protein Content (mg/g.dwt)	Sugar Content (mg/g.dwt)	ABA (ηg/g.dwt)	IAA (ηg/g.dwt)
**2850**	2.62 ± 0.32^C^	0.98 ± 0.05^C^	4.13 ± 0.52^C^	301.33 ± 27.7^C^	414.00 ± 17.9^A^
**3250**	3.55 ± 0.3^BC^	1.11 ± 0.07^C^	5.30 ± 0.35^B^	345.67 ± 16.9^C^	351.33 ± 24.3^B^
**3750**	4.26 ± 0.17^B^	1.35 ± 0.06^B^	7.51 ± 0.52^A^	418.33 ± 32.2^B^	229.00 ± 25.9^C^
**4250**	5.78 ± 0.68^A^	1.55 ± 0.06^A^	8.50 ± 0.45^A^	534.67 ± 33.9^A^	186.33 ± 7.9^C^
**LSD_(0.05)_**	**0.95**	**0.14**	**1.07**	**65.68**	**46.87**
**CV**	12.55	6.11	9.00	8.2	8.43

Means with similar letters in each column are non-significantly different according to least significant differences (LSD) (see Table 1).

**Table 3 plants-08-00378-t003:** Ecophysiological plasticity observed in *Sibbaldia procumbens* through the analysis of protein, sugar and proline, protein, and sugar contents, and ABA and IAA concentrations.

Altitude (m)	Proline Content (mg/g.dwt)	Protein Content (mg/g.dwt)	Sugar Content (mg/g.dwt)	ABA (ηg/g.dwt)	IAA (ηg/g.dwt)
**2850**	2.47 ± 0.33^C^	0.85 ± 0.03^B^	3.52 ± 0.6^C^	303.33 ± 27.7^C^	406.67 ± 16.1^A^
**3250**	3.33 ± 0.41^C^	0.95 ± 0.06^B^	4.41 ± 0.24^C^	326.00 ± 9.1^C^	369.00 ± 22.2^A^
**3750**	5.17 ± 0.33^B^	1.24 ± 0.1^A^	6.16 ± 0.19^B^	416.33 ± 16.1^B^	244.00 ± 17.1^B^
**4250**	6.58 ± 0.52^A^	1.42 ± 0.1^A^	7.41 ± 0.38^A^	482.00 ± 16.87^A^	195.67 ± 7.4^C^
**LSD_(0.05)_**	**0.94**	**0.19**	**0.89**	**43.07**	**38.24**
**CV**	11.38	9.24	8.80	5.99	6.69

Means with similar letters in each column are non-significantly different according to LSD (see Table 1).

**Table 4 plants-08-00378-t004:** Means of environmental variables measured at four different altitudinal ranges (wind speed averages are given as integers).

Environmental Variables	Altitudinal Ranges (m.a.s.l.)			
	2850	3250	3750	4250
**Latitude**	34.73111	34.71972	34.80139	34.80889
**Longitude**	73.6675	73.6525	73.66528	73.61639
**Aspect**	N	N	N	N
**Slope Angle**	40	38	38	28
**Temperature (°C)**	18.4	15.4	5.2	4.7
**Humidity**	53.85	55.3	56.9	51.2
**Heat index**	19.2	14.2	5.5	3.85
**Wind speed (m/s)**	2.5	4.5	4.5	6.5
**Dew point**	15.2	14.2	11.8	11
**Wet bulb**	14.72	15.2	12.3	13.82
**Barometric Pressure**	731	697.4	651.4	614.8
**pH**	5.9	5.3	4.7	5.1
**Electric conductivity (EC)**	1.51	1.37	3.68	2.82
**Organic matter (OM)**	1.03	1.74	1.79	1.92
**Calcium carbonate (CaCO_3_)**	7.129	5.549	6.349	7.82
**Potassium (K) (mg/kg)**	202.93	199	203	204
**Phosphorous (P) (mg/kg)**	14	14.11	12	9.3
**Sand (%)**	33.82	41	50.23	53
**Silt (%)**	42.54	27	22.76	27
**Clay (%)**	23.64	32	27.01	20

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
