# Peer review of "Ecophysiological Plasticity and Cold Stress Adaptation in Himalayan Alpine Herbs: Bistorta affinis and Sibbaldia procumbens"

_plants, 2019, doi:10.3390/plants8100378_

Round 1

Reviewer 1 Report

The title of this manuscript “Ecophysiological Plasticity and Cold Stress Adaptation in Some Himalayan Alpine Herbs” should be more specific because only two species are studied here (i.e. Sibbaldia procumbens and Bistorta affinis).

Most if not all Tables and figures are not well referenced in the text.

ABSTRACT

Line 29: this part is not discussed in the discussion section: “Induction in proline and sugar contents was found positive and significant in relation with increased elevation but negative with temperature”

INTRODUCTION

Lines 57-58: this is a larger question than the manuscript can answer. The question addresses forests, but it is not clear if forests were part of the systems that they evaluated. For instance, Fig 1 shows pastures. Climate change is not mentioned in the introduction and this is brought up in the last paragraph.

MATERIALS AND METHODS

Line 202: ‘higher moist temperate’?

Line 216: Thrice

RESULTS

Line 68: it is not ‘mentioned’ but ‘presented’.

Line 69: Figures should be presented in order, but it is first referenced ‘Fig. 1C’ and Table 2. Both seem wrongly referenced.

Line 78: Figure 1c?

Line 81: wrong table.

Figure 1: shows the same information as tables. Not sure if both are needed. ‘y-axis’ should be labelled with the metabolite measured instead of ‘compound concentration’.

Table 1 and 2: Units missing. ‘physiological variables’? be more specific. Standard errors should be added.

Table 4: Units missing. Is there a way to describe altitudinal density?

Figure 2: what do the ovoid shapes indicate? Can those axes be explained? What are the variables that drive those access and the separation of either altitude or species?

DISCUSSION

This section needs more work; otherwise about 50% of it is a repeat of the results, and authors fail to put their findings into context and how the increase in metabolites can have a major impact on plant ‘ecophysiology’

Line 156: oxygen pressure doesn’t increase with altitude.

Line 160: Authors should stop referring to ‘some plant species’ because they were two specific plant species.

Line 167: ‘adapted’

CONCLUSIONS

Line 243: why temporal? If these were not measured overtime.

Author Response

We are grateful reviewers for providing the opportunity to correct our manuscript.

The suggestions, which helped us to improve the quality of the manuscript. We accepted all modifications noted by reviewers.

On the following pages, we described the modifications in detail.

Our answers and changes are in blue, right after reviewers’ recommendations/questions, which are in black (see attachment).

Yours sincerely,

Reviewer 2 Report

Ecophysiological Plasticity and Cold Stress Adaptation in Some Himalayan Alpine Herbs

This paper deals on the ecophysiological features of some alpine plants from the Himalayas and how they adapted to such harsh environments.

Introduction. Results short and poor, not focusing the content of the paper. The objective are not clearly stated, moreover there are a very confusing sentence in such paragraph. Authors focused on phytohormones and what about the other metabolic parameters analysed?

37-38. Results very poor that authors starts defining what is Ecophisiology, that is well-known, and say nothing on plasticity or adaptation to harsh environments.

43, 167. Adapted, not adopted.

54. I don't see GA analysis in the paper so for what is it mentioned?

57-58. what is this? forest? larger spatial? Is this copy-paste?

58-61. But not only phytohormones are analysed aren't they? Explain better the objectives related to the contents of the paper.

 Results. Are too long. Results should be just a description of results without comments on explaining physiological responses that should be concentrated in the Discussion.

I don’t see the necessity to use exploratory analysis such as NMDS if anovas and GLMs do it.

Discussion. Contrarily is too short since there are sentences in results that should be incorporated to the discussion and other in Discussion that should be incorporated to objectives (158-160). Another problem is authors do not try to explain what is occurring in such environments, what are the functional outcomes of such plants to adapt to such altitudes, ecophysiology involves ecology and physiology but in the paper ecological features are not present.

Material and Methods. It is not explained why Sibbaldia and Bistorta were selected and not many other that form part of the vegetation at that altitudes. There is no any word in the paper explaining the vegetation type of such environments.

Why there are not data on 3250 m in table 4?

Although the paper could be interesting and the results are strong it still needs more work to be a real ecophysiological paper since such results. It can not be published in the present shape.

Check references there are inconsistencies. Royle, not royle. major, not mojor Colorimetric, not calorimetric.

Author Response

(The authors gave the same response as above.)

Round 2

Reviewer 2 Report

The paper was greatly improved. I do not have any objection to be published by Plants.

All the best

Rosario